# Effect of CaO, Al_2_O_3_, and MgO Supports of Ni Catalysts on the Formation of Graphite-like Carbon Species during the Boudouard Reaction and Methane Cracking

**DOI:** 10.3390/ma16083180

**Published:** 2023-04-18

**Authors:** Artem Kaporov, Oleksandr Shtyka, Radoslaw Ciesielski, Adam Kedziora, Waldemar Maniukiewicz, Malgorzata Szynkowska-Jozwik, Yelubay Madeniyet, Tomasz Maniecki

**Affiliations:** 1Institute of General and Ecological Chemistry, Faculty of Chemistry, Lodz University of Technology, 116 Zeromskiego St., 90-924 Lodz, Poland; oleksandr.shtyka@p.lodz.pl (O.S.); radoslaw.ciesielski@p.lodz.pl (R.C.); adam.kedziora@p.lodz.pl (A.K.); waldemar.maniukiewicz@p.lodz.pl (W.M.); malgorzata.szynkowska@p.lodz.pl (M.S.-J.); tomasz.maniecki@p.lodz.pl (T.M.); 2Department of Chemistry and Chemical Technology, Faculty of Chemical Technology and Natural Sciences, Toraighyrov University, 64 Lomov St, Pavlodar 140008, Kazakhstan; yelubay.m@tou.edu.kz

**Keywords:** nickel-containing catalysts, Boudouard reaction, methane cracking, nanostructured carbon allotropes, greenhouse gases transformation

## Abstract

The investigation of the course of the Boudouard reaction and methane cracking was performed over nickel catalysts based on oxides of calcium, aluminum, and magnesium. The catalytic samples were synthesized by the impregnation method. The physicochemical characteristics of the catalysts were determined using atomic adsorption spectroscopy (AAS), Brunauer–Emmett–Teller method analysis (BET), temperature-programmed desorption of ammonia and carbon dioxide (NH_3_- and CO_2_-TPD), and temperature-programmed reduction (TPR). Qualitative and quantitative identification of formed carbon deposits after the processes were carried out using total organic carbon analysis (TOC), temperature-programmed oxidation (TPO), X-ray diffraction (XRD), and scanning electron microscopy (SEM). The selected temperatures for the Boudouard reaction and methane cracking (450 and 700 °C, respectively) were found to be optimal for the successful formation of graphite-like carbon species over these catalysts. It was revealed that the activity of catalytic systems during each reaction is directly related to the number of weakly interacted nickel particles with catalyst support. Results of the given research provide insight into the mechanism of carbon deposit formation and the role of the catalyst support in this process, as well as the mechanism of the Boudouard reaction.

## 1. Introduction

It is currently recognized that greenhouse gas emissions, such as CO_2_ and CH_4_, are alarmingly high and cause serves damage to the environment. Despite international efforts to mitigate emissions of these gases, they are still emitted into the atmosphere at alarming levels that tend to increase [1].

Considerable scientific efforts have been directed towards investigating the possibility of transformations of both CO_2_ and CH_4_ into various carbonaceous materials, including carbon nanotubes, graphene, fullerenes, diamonds, hydrocarbons, carbonates, carbides, etc. These gases are an attractive source of carbon due to their cheapness, abundance, and relative safety [1,2,3,4,5,6]. The transformation process is performed using various methods, such as photocatalytic, thermal, and electrochemical methods. One specific process that utilizes these gases is the dry reforming of methane (DRM) (1), in which CH_4_ is reduced by CO_2_ to produce syngas [7].
(1)$${\mathrm{CO}}_{2}{+\mathrm{CH}}_{4}↔{2\mathrm{CO}+2H}_{2}$$

The DRM reaction can involve multiple side reactions associated with the reciprocal interactions between the reactants and products (CO_2_, CH_4_, CO, and H_2_) based on the temperature of the process (Table 1) [8,9].

**Table 1 materials-16-03180-t001:** Side reactions occurred during the DRM reaction.

Title	Equation	ΔH, kJ/mol	t, °C	No.
Reverse water–gas shift (RWGS)	${\mathrm{CO}}_{2}{+H}_{2}↔{\mathrm{CO}+H}_{2}O$	+41	200–400	(2)
Sabatier reaction	${\mathrm{CO}}_{2}{+4H}_{2}↔{\mathrm{CH}}_{4}{+2H}_{2}O$	−165	300–400	(3)
Boudouard reaction	$2\mathrm{CO}↔{\mathrm{CO}}_{2}+C$	−173	400–500	(4)
Carbon dioxide hydrogenation (CDH)	${\mathrm{CO}}_{2}{+2H}_{2}↔{C+2H}_{2}O$	−90	500	(5)
Methane cracking (MC)	${\mathrm{CH}}_{4}↔{C+2H}_{2}$	+75	600–700	(6)

During the Boudouard reaction (4) and methane cracking (6), different types of carbon deposits can be formed (Figure 1, Table 2) [10,11,12]. The type of this deposit depends on the reaction temperature and is formed via different mechanisms. It is generally accepted that during methane cracking, the first step is the dissociation adsorption of CH_4_ on metal active sites which results in the formation of elemental carbon (C_α_) and hydrogen [13]. The formed carbon can undergo slow polymerization, forming encapsulation shells (C_β_). At the same time, reactive carbon species can dissolve through metal particles and then, after reaching saturation limit, precipitate in the form of whiskers or graphite-like carbon (C_v_ or C_γ_) [14,15,16,17,18,19]. Similarly, as in the case of methane cracking, the first step in the Boudouard reaction is also the dissociative adsorption of CO on the catalyst surface [15,20,21,22]. However, to the best of our knowledge, the mechanism of the formation of carbon deposits and their type has not been investigated beyond this in the detail.

Some of the formed carbon structures can poison or mechanically destroy the catalytic system [11,23]. This occurs due to the encapsulation of the active metal site within the carbon deposit shell (Figure 1A), pore-mouth poisoning (Figure 1B), and the formation of carbon whiskers that can rip out the metal particle from the catalyst support (Figure 1C).

**Figure 1 materials-16-03180-f001:**
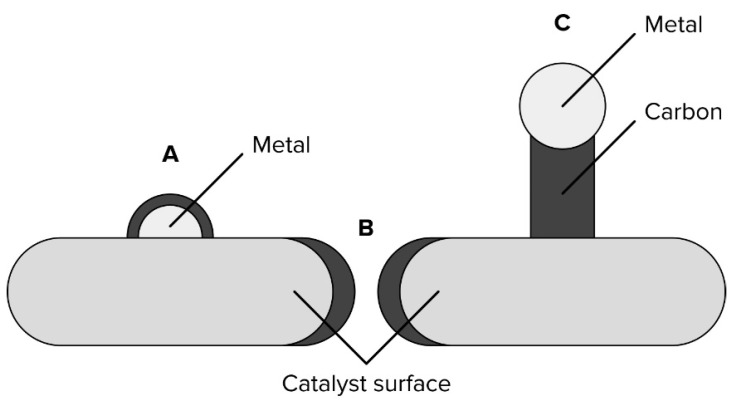
Forms of deactivation of the catalytic surface by deposited carbon: encapsulation process (**A**), pore mouth poisoning (**B**), and whisker formation (**C**).

**Table 2 materials-16-03180-t002:** Types of carbon species formed at various temperatures.

Carbon Structure	Designation	t, °C
Adsorbed, atomic carbon	C_α_	200–400
Amorphous and polymeric carbon	C_β_	400–600
Graphite-like carbon	C_γ_	500–800
Whiskers	C_v_	300–1000

Up to now, many scientific efforts have been dedicated to minimizing the formation of carbon deposits to prevent catalyst deactivation and obtain high yields of synthesis gas. However, according to the literature, Ni-based catalysts have superior properties in terms of the formation of carbon deposits when compared to iron, cobalt, and/or copper catalytic systems. In addition, it has been shown that the yield of carbonaceous materials depends on the process conditions and the chemical nature of the catalyst support [24,25,26]. Existing studies demonstrate the promising results for catalysts supported on CaO, *γ*-Al_2_O_3_, and MgO [27,28,29,30,31]. It is worth pointing out that the catalyst supports have a direct influence on the behavior of the active phase during the DRM reaction. Thus, the textural characteristics of the support phase make it possible to vary the physicochemical properties of the active sites (such as the distribution and the size of particles, etc.) [32,33]. It can be assumed that the textural characteristics as well as the acidic and basic properties of the given supports can impact the formation of graphite-like carbon species during the Boudouard reaction and methane cracking as side reactions of dry methane reforming.

In the current research, our underlying aim was to perform fundamental studies on the disposition of Ni catalysts based on various supports (namely, CaO, *γ*-Al_2_O_3_, and MgO) towards the formation of graphite-like carbon allotropes (i.e., carbon nanotubes, fullerenes, graphite, and graphene) from carbon monoxide and methane during the Boudouard reaction and methane cracking, as side reactions of dry methane reforming. The physicochemical characterization of catalytic systems and formed carbon deposits were performed using AAS, BET, TPD, TPR, TOC, TPO, XRD, and SEM.

## 2. Experimental

### 2.1. Materials and Reagents

For the preparation of the catalysts, the calcium, aluminum, and magnesium oxides (CaO, *γ*-Al_2_O_3_, MgO) by Chempur (Piekary, Poland) as catalyst supports, nickel nitrate hexahydrate (Ni(NO_3_)_2_·6H_2_O) by Chempur (Piekary, Poland) as nickel precursor, and 99% ethyl alcohol (99% EtOH) by Chempur (Piekary, Poland) as solvent were utilized. Experiments during the research were performed using mixtures of methane and carbon monoxide in argon (25% CH_4_/Ar, 25% CO/Ar) by Linde (Warsaw, Poland) as reaction gases.

### 2.2. Catalyst Preparation and Experimental Measurements

Ni catalysts based on CaO, *γ*-Al_2_O_3_, and MgO were prepared by the impregnation method from ethanol solutions of nickel nitrate and oxides used under magnetic stirring. Precipitates obtained were evaporated from the suspensions, dried, and calcined at 500 °C for 4 h. The content of the active phase in each catalytic system was about 20% wt.

The Boudouard reaction and methane cracking were conducted in a continuous flow reactor (Figure 2) at 450 and 700 °C, respectively. The amount of the catalyst for each experiment was 0.5 g. Reaction gases were utilized with a flow rate of 50 mL/min. Prior to the measurements, samples were reduced in situ in a hydrogen atmosphere (5% H_2_/Ar) at 500 °C for 2 h.

### 2.3. Analysis of Catalysts and Products

The gas composition during the reactions was measured using Chromatron GCHF 18.3 (Berlin, Germany) and INCO 505M (Warsaw, Poland) online gas chromatographs with thermal conductivity detectors (TCD). The conversion of CO and CH_4_ was calculated using the following formula:$$C_{g}=\frac{W_{g}-\left(W_{g}^{\prime }\times \frac{W_{\mathrm{Ar}}}{W_{\mathrm{Ar}}^{\prime }}\right)}{W_{g}}\times 100\%,$$
where, C_g_—used gas conversion, %; W_g_—used gas content measured before reaction; W_Ar_—argon content measured before reaction; W′_g_—used gas content measured during the reaction; W′_Ar_—argon content measured during the reaction.

Physicochemical characterization of synthesized catalytic systems was performed using atomic adsorption spectroscopy (AAS), Brunauer–Emmett–Teller method analysis (BET), temperature-programmed desorption of NH_3_ and CO_2_ (NH_3_- and CO_2_-TPD), and temperature-programmed reduction (TPR).

The atomic adsorption spectroscopy was carried out using a Unicam Atomic Adsorption SOLAAR M6 spectrometer (Cambridge, MA, USA) at a wavelength of 232.0 nm in an acetylene/air flame. The samples were weighed on analytical scales, 3 mL of 65% nitric acid was added to each sample, and they were subjected to initial mineralization in a BANDELIN ultrasonic bath (Berlin, Germany) at 25 °C for 25 min. Then, 1 mL of 65% nitric acid was added to each sample and the samples were mineralized in a Milestone UltraWave closed microwave system (Rome, Italy). The dissolved samples were placed in 100 mL volumetric flasks and deionized water was added. The blank solution underwent the same procedure.

The Brunauer–Emmett–Teller method studies were performed on a Micrometrics AutoChem II+ instrument (Ottawa, ON, Canada). A sample weighing 0.2 g was placed in a fluidized bed reactor and purged in a helium flow at a rate of 50 mL/min to 350 °C at 50 °C/min, followed by cooling of the reactor in an air flow to ambient temperature. Physisorption of the analyzed gas (26% N_2_/He) was conducted using a cryogenic bath to –190 °C and further desorption using a water bath to ambient temperature.

The temperature-programmed desorption of NH_3_ and CO_2_ was also conducted using a Micrometrics AutoChem II+ apparatus (Ottawa, ON, Canada). Before TPD experiments, the samples (0.2 g) were purged with helium at 600 °C (for NH_3_ sorption) and 800 °C (for CO_2_ sorption) for 1 h to remove any contaminations. After cleaning, the samples were cooled and saturated for 30 min in a flow of pure NH_3_ or CO_2_ at 50 °C. In both cases, the total flow rate was 25 mL/min. Then, the samples were purged in helium flow at 100 °C until a constant baseline level was attained. TPD measurements were performed in the temperature range 100–600 °C (NH_3_ sorption) and 100–800 °C (CO_2_ sorption) at a rate of 20 °C/min using helium as carrier flow. The evolved NH_3_ or CO_2_ was detected by an online thermal conductivity detector (TCD) calibrated by the peak area of known pulses of NH_3_ or CO_2_.

The temperature-programmed reduction was carried out by a Micrometrics AutoChem II+ device (Ottawa, ON, Canada). The samples (0.1 g) were pre-heated in an argon flow at 300 °C for 20 min. After that, the reactor was cooled to 40 °C. Measurement of the reduction rate was performed in hydrogen flow (5% H_2_/Ar) with a rate of 20 mL/min to 800 °C at 10 °C/min. The hydrogen uptake was detected by a thermal conductivity detector (TCD).

Qualitative and quantitative assessments of carbon deposits formed after the processes were carried out using total organic carbon analysis (TOC), temperature-programmed oxidation (TPO), and X-ray diffraction (XRD).

The total organic carbon analysis was conducted using a Shimadzu TOC-5000 with an SSM-5000 solid sampling module (Tokyo, Japan). To achieve this, 0.2 g of the sample was placed in a ceramic boat and placed in the SSM sampling unit. The measurements were performed by burning the sample in an oxygen atmosphere at 900 °C.

The temperature-programmed oxidation was carried out in a quartz reactor in the temperature range from 25 to 800 °C with a linear heating rate of 10 °C/min. Carbonized samples (weight about 0.1 g) were oxidized in an oxygen (5% O_2_/Ar) stream with a volumetric flow rate of 40 mL/min. The evolution of the gaseous products was analyzed as a function of temperature using a Hiden Analytical HPR20 mass spectrometer (London, UK).

Room temperature X-ray powder diffraction patterns were collected using a PANalytical X’Pert Pro MPD diffractometer (London, UK) in Bragg-Brentano reflection geometry. The diffractometer was equipped with Cu K_α_ radiation source (λ = 1.5418 Å). Data were collected in the 2θ range of 5–90° with a step size of 0.0167° and exposure per step of 27 s. Because the raw diffraction data contained some noise, the background during the analysis was subtracted using the Sonneveld, E. J., and Visser algorithm. The data were then smoothed using a cubic polynomial function.

The scanning electron microscopy measurements were performed using a scanning electron microscope HITACHI (Tokyo, Japan), equipped with an energy dispersive spectrometer EDS by Thermo Noran (New York, NY, USA).

## 3. Results and Discussion

### 3.1. Characterization of Catalysts

Table 3 demonstrates the physicochemical properties of used catalysts. The specific surface area (S_BET_) varied widely depending on the catalyst support. The highest specific surface area of 91 m^2^/g was observed for Ni/*γ*-Al_2_O_3_ catalyst while the lowest (about 7 m^2^/g) was for Ni/CaO. The surface area of Ni/MgO was between these extremes and was equal to 51 m^2^/g. Moreover, the specific surface area can indirectly reveal the dispersion of nickel particles on the catalyst surface [34]. Corresponding to the fact that the Ni/*γ*-Al_2_O_3_ catalyst demonstrated the largest surface area, it can be predicted that the Ni dispersion will also be the highest. At the same time, the lowest S_BET_ was observed for the Ni/CaO sample and, accordingly, this sample is expected to show the worst dispersion of Ni particles.

The highest number of acidic sites was observed for Ni/*γ*-Al_2_O_3_ and Ni/MgO; the ammonia uptakes were 0.5 and 0.3 mmol/g, respectively. In addition, these catalysts were characterized by almost uniform distribution in the strength of acidic sites, i.e., the contribution of weak, medium, and strong sites to the total acidic sites was almost equal. The lowest amount of ammonia uptake (0.2 mmol/g) was observed in the case of Ni/CaO. Further, this catalyst was characterized by the highest amount of strong acidic sites—87% of total sites.

The highest number of basic sites was determined for Ni/MgO (22.1 mmol/g), while the concentration of other catalytic systems was three times lower than that of the basic sites, averaging about 6.7 mmol/g. The distribution of these sites is shifted towards medium and strong sites in the case of Ni/*γ*-Al_2_O_3_ and towards weak and medium strength sites in the case of Ni/MgO catalyst. On the other hand, the Ni/CaO catalytic system had the highest contribution of strong sites, 64% of total basic sites.

**Table 3 materials-16-03180-t003:** Physicochemical properties of the synthesized catalysts.

Catalyst	Ni Content, % wt.	Pore Volume, cm^3^/g	S_BET_, m^2^/g	Acidity, mmol/g				Basicity, mmol/g			
				Weak 100–300 °C	Medium 300–450 °C	Strong 450+ °C	Total	Weak 100–300 °C	Medium 300–450 °C	Strong 450+ °C	Total
Ni/CaO	22.3	0.04	7	>0.1	>0.1	0.2	0.2	0.7	1.3	3.6	5.6
Ni/*γ*-Al_2_O_3_	21.8	0.29	91	0.2	0.2	0.1	0.5	1.5	3.3	3.0	7.8
Ni/MgO	20.3	0.16	51	0.1	0.1	0.1	0.3	9.7	9.2	3.2	22.1

Figure 3 and Table 4 show the temperature-programmed reduction studies of the catalysts used. Ni/CaO catalyst (Figure 3A) is characterized by two main peaks—the first peak at 520 °C (peak α) is ascribed to free NiO species with a weak interaction with CaO, and the second at 640 °C (peak CaO) represents the different carbonate structures formed over the catalyst support phase [35].

For the Ni/*γ*-Al_2_O_3_ pattern (Figure 3B), three main peaks were determined. The first peak at 420 °C (peak α) is assigned to free NiO species with a weak interaction with the *γ*-Al_2_O_3_ phase, and the second at 590 °C (peak β) represents the NiO species in a strong interaction with alumina, which can also be identified as NiO species with mixed oxide phases. The third peak at 740 °C (peak γ) is related to NiO species that are ascribed to the stable NiAl_2_O_4_ phase with spinel structure [36].

The TPR profile of the Ni/MgO (Figure 3C) sample is marked by four peaks. The first peak at 320 °C (peak α) is ascribed to non-stochiometric Ni_2_O_3_, however, the occurrence of this peak is correlated to black nickel(III) oxide against the progressive pale green NiO. The second peak at 430 °C (peak β) is assumed to represent the reduction in “unreacted” NiO located on the surface of the catalyst support phase. The third peak at 510 °C (peak γ) can be assigned to some form of Ni^2+^ ions having square pyramidal coordination in the outermost layer of the MgO structure. The last peak, assumably at 840 °C (peak δ), indicates the reduction of NiO forms located in the subsurface layers of the MgO lattice [37].

It is worth noting that the degree of reduction of NiO is virtually equal for Ni/CaO and Ni/*γ*-Al_2_O_3_ (93 and 88%, respectively), while for Ni/MgO it is much lower (55%). Therefore, it becomes clear that the reaction surface area of Ni/MgO is much lower than Ni/CaO and Ni/*γ*-Al_2_O_3_, from which it can be assumed that both the Boudouard reaction and methane cracking over this catalyst will proceed to be much less active and, accordingly, the yield of products will also be lower.

**Figure 3 materials-16-03180-f003:**
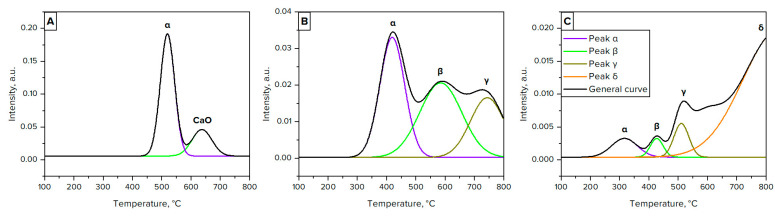
Temperature-programmed reduction profiles for Ni/CaO (**A**), Ni/*γ*-Al_2_O_3_ (**B**), and Ni/MgO (**C**).

**Table 4 materials-16-03180-t004:** Temperature-programmed reduction of Ni catalysts.

Sample	Peaks								NiO Reduction, %
	α		β		γ		δ		
	t, °C	Area	t, °C	Area	t, °C	Area	t, °C	Area	
Ni/CaO	520	11.3	640 *	3.5 *	–	–	–	–	93
Ni/*γ*-Al_2_O_3_	420	3.5	590	3.7	740	2.4	–	–	88
Ni/MgO	320	0.3	430	0.2	510	0.3	840	6.0	58

* Peak β for Ni/CaO catalyst is related to CaO.

### 3.2. Reaction Studies

Figure 4 shows the change in the conversion of the gases during the processes. Ni/CaO and Ni/*γ*-Al_2_O_3_ demonstrated superior activity during the methane cracking than Ni/MgO. This is evidenced by the high conversion rate of CH_4_ (about 90%) over these catalysts at the beginning of the experiments (Figure 4A,B), which is directly confirmed by obtained TPR profiles of the catalytic samples. The rapid decrease in the conversion rate of CH_4_ (from 70% to 10% in 15 min) over Ni/MgO catalysts is explained by the low amount of nickel particles in metal form (Table 4). On the other hand, Ni/CaO and Ni/*γ*-Al_2_O_3_, which have in their composition superficial and weakly bound Ni particles, have smooth and gradually decreasing values of CH_4_ conversion rate (from 80–90% to 15% in 45 min). The changes in CO conversion, similar to methane cracking, depend on the degree of reduction in Ni particles in the catalyst structure. This is evidenced by the transformation of CO over Ni/MgO, which was about 1.5 times lower than for Ni/CaO and Ni/*γ*-Al_2_O_3_. All catalytic systems demonstrated a similar decrease in activity with time on stream.

### 3.3. Analysis of Products

The decrease in the conversion of reaction gases is directly associated with the formation of carbon deposits on the surface of active metal sites. Table 5 and Figure 5 give the quantitative and qualitative assessment of formed carbon species. The greatest amount of carbon deposits was observed on the surface of most active catalysts. More particularly, about 18–22% wt. of carbon was formed on Ni/CaO and 15–23% wt. on Ni/*γ*-Al_2_O_3_ catalyst. Meanwhile, only 6–8% wt. of carbon deposit was formed on the surface of the least active catalytic system—Ni/MgO.

**Table 5 materials-16-03180-t005:** Carbon content in the samples after the experiments.

Sample	Carbon Content, % wt.	
	Boudouard Reaction	Methane Cracking
Ni/CaO	18.6	21.7
Ni/*γ*-Al_2_O_3_	22.9	15.0
Ni/MgO	8.6	6.3

The XRD analysis results revealed that all investigated catalysts tended to sinter when exposed to high temperature. In particular, on average, the size of nickel crystallites increased twofold after the investigated processes (Table 6). It is crucial to note that the assumptions regarding the dependence of the dispersion of Ni particles on the specific surface area of catalysts was affirmed. Thus, the sample with the lowest specific surface area was characterized by the poorest dispersion of nickel. Meantime, the best dispersion was observed for Ni/*γ*-Al_2_O_3_ with the highest S_BET_.

**Table 6 materials-16-03180-t006:** Ni crystallite size before and after the experiments.

Sample	Ni Crystallite Size, nm		
	Reduced Catalysts	Boudouard Reaction	Methane Cracking
Ni/CaO	24	34	56
Ni/*γ*-Al_2_O_3_	11	26	25
Ni/MgO	15	26	29

On the other hand, the dispersion of Ni particles does not correlate with the activity and stability of the catalysts in the investigated reactions. For instance, Ni/CaO and Ni/γ-Al_2_O_3_ samples exhibit a relatively similar yield of carbon deposit and catalytic activity (Figure 4, Table 5), although the latter catalyst was characterized by the best dispersion of Ni particles among investigated samples.

The results of TPO measurements (Figure 5) are in good agreement with those of TOC (Table 5). The oxidation profiles of investigated catalysts demonstrated a broad peak in the temperature range from 400 to 700 °C. This effect can be attributed to the oxidation of graphite-like carbon species, particularly carbon filaments (which are formed at temperatures of about 500 °C) and carbon nanotubes (which are observed at temperatures of about 700 °C) [38]. These two types of carbon structures are clearly visible on the TPO pattern of Ni/MgO (Figure 5E,F) where two distinctive peaks with the maxima at 550 °C and 650–700 °C can be observed. It is worth noting that the aforementioned carbon allotropes can mechanically destroy the catalyst by ripping metal particles off the support surface. This can be expected in the case of nickel catalysts supported on CaO and *γ*-Al_2_O_3_ where most of the metal particles are represented as weakly bonded species. Therefore, the deactivation of these catalysts is not as rapid as for Ni/MgO, where metal particles are strongly bonded with a support surface. For this occasion, nickel particles are likely to remain on the surface and become coated with the forming carbon deposit. This leads to the rapid deactivation of Ni/MgO, as evidenced by the loss of its activity in the investigated processes (Figure 4).

**Figure 5 materials-16-03180-f005:**
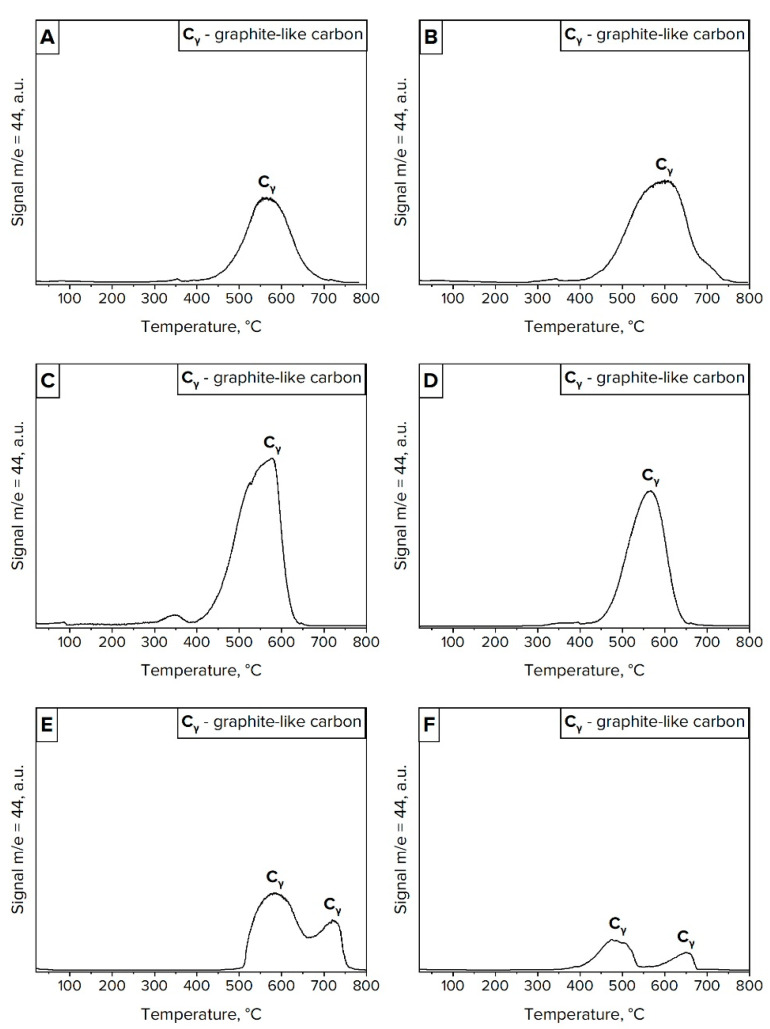
Temperature-programmed oxidation analysis of Ni/CaO samples after the Boudouard reaction (**A**) and methane cracking (**B**), Ni/*γ*-Al_2_O_3_ after the Boudouard reaction (**C**) and methane cracking (**D**), and Ni/MgO after the Boudouard reaction (**E**) and methane cracking (**F**).

The TPO measurements for each catalyst showed similar oxidation profiles for both the Boudouard reaction and methane cracking processes. In this regard, it can be concluded that the formation of graphite-like carbon species followed the same mechanism irrespective of the reaction gases. Particularly, the first step is the dissociative adsorption of CO on the active metal sites, which led to the formation of elemental carbon. This is followed by the dissolution of carbon in metal particles and precipitation in the form of filaments and/or carbon nanotubes.

The X-ray diffraction patterns of both fresh catalysts and samples after the Boudouard reaction and methane cracking are presented in Figure 6. The Ni/CaO diffraction pattern has peaks related to the CaCO_3_ phase at 2θ of 29°, to the CaO phase at 2θ of 33°, 37°, and 54°, and to the Ni phase at 2θ of 44° (Figure 6A). Ni/*γ*-Al_2_O_3_ and Ni/MgO profiles demonstrate peaks related only to the Ni phase at 2θ of 44° and 52° (Figure 6B), and at 2θ of 36° and 44° (Figure 6C), respectively. Most of the deposited carbon species related to peak at 2θ of 26° and were represented in the form of 2H graphite identified as the crystallographic form of nanostructured carbon materials. Meanwhile, for Ni/CaO after the Boudouard reaction, only a CaCO_3_ peak was observed (Figure 6A). This phenomenon is associated with the sorption capacity and chemical properties of CaO, due to which it is not distinguished for methane cracking, as well as for other catalytic samples [39].

The foregoing XRD and TPO results are confirmed by SEM (Figure 7). The microscope images give us an affirmation that most of the carbon structures obtained during the reactions studied are filaments, i.e., graphite-like carbon species. The most prominent are the products obtained using Ni/*γ*-Al_2_O_3_ as well as Ni/CaO with an average length of about 4–10 µm, while for Ni/MgO the length is less than 2–4 µm.

## 4. Conclusions

The effect of CaO, *γ*-Al_2_O_3_, and MgO supports of Ni catalysts on the formation of carbon deposits during the Boudouard reaction and methane cracking was established. It was found that irrespective of the process, graphite-like carbon species were formed. Their formation occurred on the reduced nickel particles and, therefore, the amount of carbon depended on the degree of reduction in catalysts. The order of reducibility of catalytic samples can be represented as follows: Ni/CaO → Ni/*γ*-Al_2_O_3_ → Ni/MgO. The difficult reducibility of Ni species on the surface *γ*-Al_2_O_3_ and MgO was explained by the migration of Ni^2+^ ions into the structure of the support. Therefore, the most carbonized catalyst was Ni/CaO since it had the highest amount of Ni species weakly bound to the catalyst support. The formed carbon deposits were presented on the surface of the catalysts, mainly in the form of filaments.

## Figures and Tables

**Figure 2 materials-16-03180-f002:**
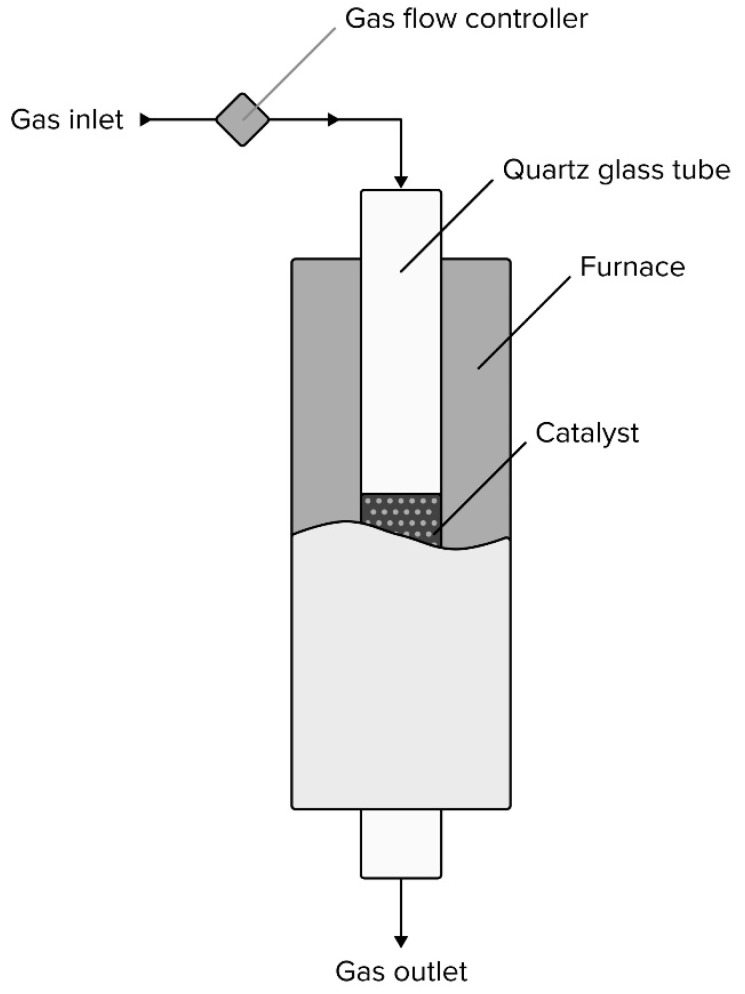
Schematic representation of the reactor for the process.

**Figure 4 materials-16-03180-f004:**
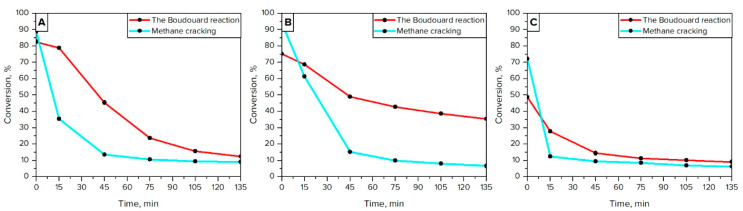
Conversion of CO and CH_4_ within the reactions using Ni/CaO (**A**), Ni/*γ*-Al_2_O_3_ (**B**), and Ni/MgO (**C**).

**Figure 6 materials-16-03180-f006:**
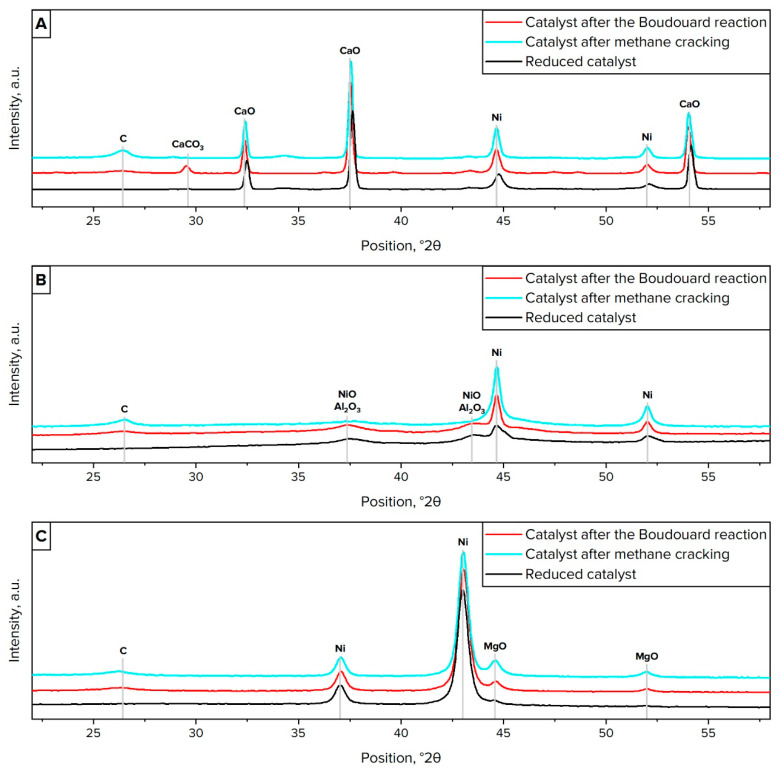
X-ray diffraction analysis of Ni/CaO samples (**A**), Ni/*γ*-Al_2_O_3_ samples (**B**), and Ni/MgO samples (**C**).

**Figure 7 materials-16-03180-f007:**
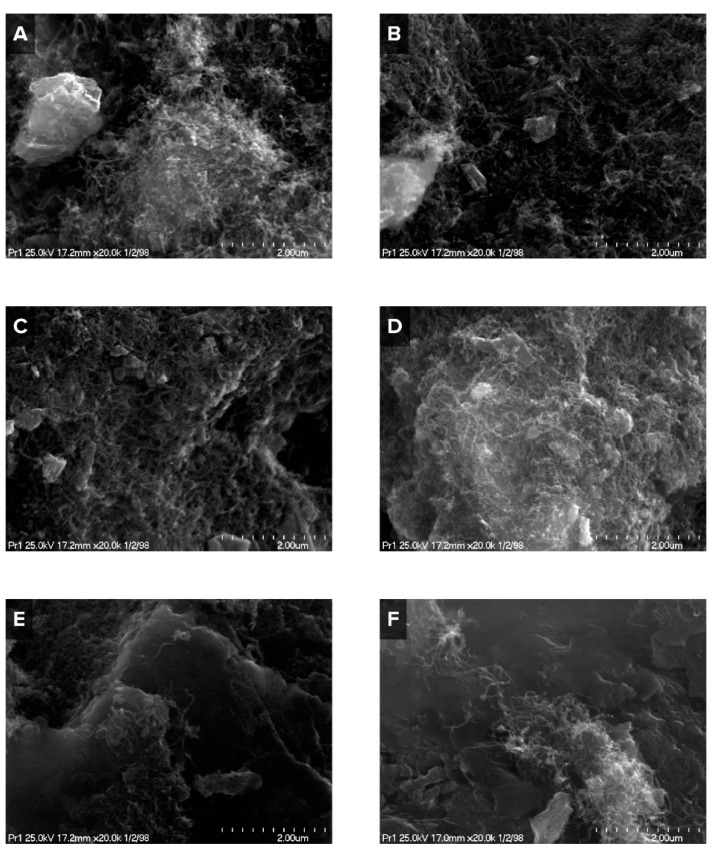
Scanning electron microscopy images of Ni/CaO samples after the Boudouard reaction (**A**) and methane cracking (**B**); Ni/*γ*-Al_2_O_3_ after the Boudouard reaction (**C**) and methane cracking (**D**), and Ni/MgO after the Boudouard reaction (**E**) and methane cracking (**F**).

## Data Availability

Data will be made available on request.

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
