# Peer review of "Effect of CaO, Al2O3, and MgO Supports of Ni Catalysts on the Formation of Graphite-like Carbon Species during the Boudouard Reaction and Methane Cracking"

_materials, 2023, doi:10.3390/ma16083180_

Round 1

Reviewer 1 Report

The paper investigated the role of the support on the activity of Ni catalysts for Boudouard reaction and methane cracking. Three different supports were compared; these supports have very different surface areas and acidic/basic properties. Thus, it is difficult to analyze each effect individually. Ni dispersion must be measured to evaluate the effect of Ni particle size on carbon formation. Moreover, the authors should improve the discussion and comparison with other works in the literature.

I have specific comments/suggestions for the authors:

1. Page 3. Why were ethanol solutions of nickel nitrate used for support impregnation (instead of aqueous solutions)?

2. Page 4: “Prior to the measurements, samples were reduced in situ in a hydrogen atmosphere (5% H2/Ar) at 500 °C for 2 h.” This reduction temperature must be justified because TPR analysis (Figure 3) showed that Ni/Al2O3 and Ni/MgO are reduced at higher temperatures.

3. Page 5. The number of acidic sites in the text does not match the number displayed in Table 3. The number of acidic and basic sites should be compared with other works in the literature. If Ni/CaO catalyst has the lowest number of acid sites, why does not this catalyst have the highest number of basic sites? The basicity of CaO is lower than Al2O3 on a weight basis (mmol/g) but it is much higher on a surface basis (mmol/m2); this must be discussed in the text.

4. Page 6: “It is worth noting that the degree of reduction of NiO is virtually equal for Ni/CaO and Ni/Al2O3 (93 and 88%, respectively), while for Ni/MgO it is much lower (55%). Therefore, it becomes clear that the reaction surface area of Ni/MgO is much lower than Ni/CaO and Ni/Al2O3”.  The lower reduction degree does not necessarily indicate lower metal surface area available for reaction, because it depends on Ni dispersion. The surface area of MgO is much higher than that of CaO, and it can provide a better dispersion for Ni particles. It is fundamental to measure Ni dispersion from Ni crystallite sizes (using XRD patterns of the reduced catalysts) or from H2 chemisorption.

5. Figure 4: The experimental points must be shown.

6. Page 7: “Moreover, it should be noted that Ni/MgO is characterized by the greatest basicity than other catalysts (22.1 mmol/g). This fact makes it possible to confirm earlier assumptions about the reactivity of Ni/MgO, as well as to argue that the reaction of methane cracking is directly dependent on the strength of basic sites and the number of reduced Ni particles.” What is the role of basic sites on methane activation? Methane activation is probably done only by metal sites.

7. Pages 8-9. TPO analysis provides an indication of the nature of the carbon formed, but it is essential to show SEM images to confirm the formation of graphite-like carbon species (filaments or nanotubes).

8. Page 9. “…the first step is the dissociative absorption of CO on the active sites, which led to the formation of elemental carbon.” Which active sites, metal sites, or acidic/basic sites? This needs to be discussed more in-depth.

9. The main conclusion “…for the most effective transformation of carbon monoxide and methane into nanostructured carbon allotropes, it is necessary to select the catalytic systems with a weak interaction between the active phase and support surface.” is very fragile. The difference in catalyst activity is related to several effects (Ni dispersion, acidic/basic properties, reduction degree), not just the strength of metal-support interaction.

Author Response

Dear Reviewer,

On behalf of me and my co-authors, I am grateful for your helpful and appreciated remarks on our paper. We have included the necessary revisions as well as made some new analyses and calculations and would like to respond to your comments:

1. The continuation of the given research implies the application of various hydrophobic materials as the catalysts’ components. We would like to maintain the convergence of the results for comparison; therefore, it was decided to synthesize the investigated catalytic systems using ethanol as a solvent.

2. The selection of the temperature of the catalysts’ reduction was based on the possible physical transformations of the sample under investigation. We agree that the reduction temperature can be higher, however, it should be noted that the sintering of catalytic materials can occur at temperatures higher than 600-700 °C. Furthermore, because the TPR analysis, contrary to the reduction process in this paper, is not isothermal, the results do not fully represent the behavior of the catalyst in the reduction at a certain temperature. In the case of isothermal reduction, NiO will be virtually completely reduced to metallic Ni. In view of this, we believe that 500 °C is the most optimal for the reduction of most of the nickel particles in the catalyst structure and also to avoid sintering, which is equally important.

3. The number of acidic sites in the text after our re-validation has been corrected and is now complete with Table 3. Unfortunately, we cannot compare the number of acidic and basic sites between themselves for one sample. If we use different probe molecules (for instance, pyridine instead of ammonia), we will obtain a completely different number of the basic sites for our samples. Therefore, this technique is suitable only to observe changes in the number of both acidic and basic sites separately for investigated samples. For this reason, we also could not make a direct comparison with the literature as the TPD measurement conditions differ.

In all our experiments we use the same weight of catalysts. Consequently, all obtained values (including BET) were expressed per unit of a gram of sample. In our opinion, there is no necessity to recalculate the acid-base values to mmol/m2. In this case, to make a straightforward comparison, we would need to recalculate the catalytic values as well.

4. Thank you for your note. We have calculated the nickel crystallite sizes in the examined catalysts both before and after the reaction. The missing information has been added to the manuscript (page 11, Table 6).

5. The experimental points have been marked and shown for each graph (page 9, Figure 4).

6. We have considered methane cracking in the context of dry methane reforming, in the mechanism of which the basic sites have a significant role. However, you are completely right that in the particular reaction, the methane activation is proceed only on the metal sites. Our sentence was not formulated clearly, and we have corrected this mistake (page 9). Thank you for this valuable remark.

7. We have confirmed the formation of carbon filament species using SEM. The corresponding images have been added to the manuscript (pages 13-14, Figure 7).

8. Thank you for your appreciated remark. We have clarified in the text using the literature which sites take place in the adsorption of CO (page 12).

9. We have reformulated our conclusions in accordance with your suggestions and analyses carried out (page 14).

We trust that our paper is now completely compliant with your recommendations and that it will be instructive and suitable for the readers of the Journal.

With gratitude and appreciation,
Artem Kaporov.

Reviewer 2 Report

The current manuscript discussed Effect of CaO, Al2O3, and MgO supports of Ni catalysts on the formation of graphite-like carbon species during the Bou-douard reaction and methane cracking. The manuscript is well written, the concepts are explained well but there are some concerns which needs to be addressed before considering the manuscript for publication.

There are some comments:

1.     Figure 2 the base of reactor is not labelled.

2.     The composition of Ni content in table 3 is selected on what bases? Can authors explain it with literature support.

3.     The concepts of temperature-programmed desorption and reduction should be elaborated more in detail. Why these studies for bare Ni catalyst are not performed.?

4.     The peaks in figure 3 alpha, beta and gamma etc needs to be explained well in the discussion. They are also not mentioned in the captions.

5.     In figure 4, Conversion of CO and CH4 within the reactions using Ni/CaO , Ni/Al2O3 , and Ni/MgO but no data for Ni catalyst is provided in any comparison in the manuscript. For a better comparison authors should provide the conversion efficiency of bare Ni catalyst. Same applies to Figures 5 and 6(XRD comparison).

6.     In addition, no morphology studies are provided. It will be helpful to perform and provide SEM and XRD data for pre and post samples from Bou-douard reaction and methane cracking, and observe what changes took place on the catalyst before and after the reaction.

Author Response

Dear Reviewer,

On behalf of me and my co-authors, I am grateful for your helpful and appreciated remarks on our paper. We have included the necessary revisions as well as made some new analyses and calculations and would like to respond to your comments:

1. Unfortunately, we did not quite understand your remark.

2. This nickel content in the catalyst composition of 20% wt. is the most commonly used metal loading in the scientific literature for the dry reforming of methane. In the future, we planned to use our results for methane cracking and the Boudouard reaction to explain the course of dry methane reforming. Therefore, in order to be able to make a comparison with the scientific literature we have to use the same ratio of active component and catalyst support. Additionally, our previous experiments have shown that the amount of carbon deposit formed on the catalyst surface with lower metal loading (2 and 5% wt.) is low and not sufficient to perform characterization measurements.

https://palotina.ufpr.br/catprobio/wp-content/uploads/sites/7/2016/11/Methane-dry-reforming-using-Ni-Al2O3-catalysts-Evaluation-of-the-Carine-A-Schwengber.pdf

https://www.sciencedirect.com/science/article/abs/pii/S1875510015301980?via%3Dihub

https://www.sciencedirect.com/science/article/abs/pii/S1875510015301980?via%3Dihub

3 and 5. The given research was performed to identify the correlation between various supports and the formation of nanostructured carbon allotropes and, proceeding from that, we believe that the studies you proposed can be used in a project aimed at investigation of the active phases, which will be a logical continuation of this article. If you refer to the bulk Ni catalyst, it would be inactive in the investigated processes due to a low active surface.

4. Thank you for your remark. We have added the peak titles to the description of the TPR results (page 8).

6. For the morphological characterization of the samples both after and before the reactions, X-ray diffraction studies were performed. According to the obtained data, the morphology of the catalysts, as well as their phase composition were established. The Ni crystallite size calculated from the XRD data has been added to the text (page 11, Table 6) as well as SEM analysis has been carried out and results have been appended to the manuscript (pages 13-14, Figure 7).

We trust that our paper is now completely compliant with your recommendations and that it will be instructive and suitable for the readers of the Journal.

With gratitude and appreciation,
Artem Kaporov.

Reviewer 3 Report

Comments are attached.

Author Response

Dear Reviewer,

On behalf of me and my co-authors, I am grateful for your helpful and appreciated remarks on our paper. We have included the necessary revisions as well as made some new analyses and calculations and would like to respond to your comments:

1. We have performed pore size measurements (page 8, Table 3) for the samples before the reactions. Unfortunately, however, we could not complete the characterization of the used catalysts' surface morphology before the revision period's end. Based on the XRD measurements, we have observed the sintering of metal particles during the reactions, hence we can assume that the pore size and specific surface area after the experiments performed will be smaller than before.

2. It is rather complicated to accurately assess and compare the conversion results obtained with other studies. In the majority of papers conversion is reported only for its maximum level; however, in our article, we have performed measurements throughout the experiment, and it would seem to us that it would not be quite appropriate to correlate the results.

https://www.mdpi.com/2073-4344/13/4/647

https://www.sciencedirect.com/science/article/abs/pii/S0959652620323039

https://www.beilstein-journals.org/bjnano/articles/9/108

3. We have reformulated our conclusions in accordance with analyses carried out and appended new information (page 14). 

We trust that our paper is now completely compliant with your recommendations and that it will be instructive and suitable for the readers of the Journal.

With gratitude and appreciation,
Artem Kaporov.

Reviewer 4 Report

Please, see attached file

Author Response

Dear Reviewer,

On behalf of me and my co-authors, I am grateful for your helpful and appreciated remarks on our paper. We have included the necessary revisions as well as made some new analyses and calculations and would like to respond to your comments:

1. We have considered the methane cracking and the Boudouard reaction in the context of dry methane reforming, in the mechanism of which the acidic and basic sites have a significant role. However, in the particular reaction of the methane and carbon monoxide activation is proceed only on the metal sites. 

2. We have redacted the purpose and objectives of the research (page 3).

3. We have made the structure of the “Experimental” section similar to the one you wrote (pages 3-6).

4. We have included information about the reagents used (page 3) and the phase of the aluminum oxide (gamma) under study throughout the text.

5. We have made the structure of the “Results and Discussion” section similar to the one you wrote (pages 6-14).

6. We have corrected these deficiencies (page 8, Table 3) and also have appended the pore size data (page 8, Table 3).

7. You are absolutely right that the effect of the catalyst support also depends on the properties of the metal particles on its surface. However, because of the extensive volume of this paper, this aspect has not been investigated, but we intend to do that in our next project. Thank you for this valuable remark.

We trust that our paper is now completely compliant with your recommendations and that it will be instructive and suitable for the readers of the Journal.

With gratitude and appreciation,
Artem Kaporov.

Round 2

Reviewer 1 Report

Most corrections were made, but some comments were not adequately responded to.

Author Response

Dear Reviewer,

We are grateful for your comments. Unfortunately, we do not fully agree with you and would like to answer your questions and remarks.

3. The number of acidic sites in the text still does not match the number displayed in Table 3. The highest number of acidic sites was observed for Ni/ -Al2O3 and Ni/MgO. The ammonia uptakes were 0.50 and 0.30 mmol/g, respectively. Also, these catalysts were characterized by almost uniform distribution in the strength of acidic sites, i.e., the contribution of weak, medium, and strong sites to the total acidic sites was almost equal. The lowest amount of ammonia uptake (0.20 mmol/g) was observed in the case of Ni/CaO. The values in Table 3 are: 0.5, 0.4 and 0.3 mmol/g for Ni/ -Al2O3, Ni/CaO and Ni/MgO, respectively. Thus, in Table 3, the lowest value is for Ni/MgO (0.3 mmol/g) and not for Ni/CaO.

We profoundly apologize for this inaccuracy. The table now indicates the correct value (Table 3, page 8).

In relation to we also could not make a direct comparison with the literature as the TPD measurement conditions differ. Although TPD measurement conditions may differ from one work to another, it is still possible to compare values (see, for example, Applied Catalysis B: Environmental 264 (2020) 118494 and International Journal of Hydrogen Energy 45 (2020) 22906).

In relation to “In all our experiments we use the same weight of catalysts. Consequently, all obtained values (including BET) were expressed per unit of a gram of sample. In our opinion, there is no necessity to recalculate the acid-base values to mmol/m2.” The density of acid/basic sites on surface basis (mmol/m2) is essential to compare catalysts that have very different surface areas, as in this case (BET areas from 7 to 91 m2 /g). A lower number of basic sites per gram does not necessarily means a lower concentration of basic sites available on the support surface. There is no need to recalculate the catalytic values.

Unfortunately, we do not agree with you that a comparison of the measured acid-base properties with the literature would be possible and meaningful. We have three different catalytic supports with a particular metal loading. It is almost impossible to find similar catalytic systems in a single publication. We concur that it is possible to find similar catalytic systems in different papers, however, it will not be meaningful due to the various measurement conditions. Furthermore, our study was not aimed at changing the acidity and basicity as a result of catalyst modification and/or preparation; such measurements were made only to exhibit the fundamental difference between investigated catalytic systems. A comparison with the literature would only demonstrate whether we obtained similar or different acid-base properties.

We can express the acid-base properties in "mmol/m2" to demonstrate the concentration of the acidic and basic sited available on the surface. In our opinion, in this case, it would be correct to express also the activity of the catalyst per surface area/active sites (thus a comparison of these two variables would be more meaningful). Nonetheless, in the course of our investigation, we concluded that the acid-base properties did not impact the examined processes. Therefore, we believe that recalculating these values would not lead to any new findings.

6. The authors answered that “the methane activation is proceed only on the metal sites. Our sentence was not formulated clearly, and we have corrected this mistake (page 9).” The authors only removed the sentence from the text, but they did not correlate the activity for methane cracking with metal dispersion (Table 6 added to the revised version of the manuscript).

9. The sentence that was added to the conclusion “As opposed to the catalyst reducibility, the dispersion of nickel particles on the support surface was a secondary factor impacting the examined processes.” is not clear. What is the effect of metal dispersion on the activity for methane cracking and Boudouard reaction? Were catalysts with higher Ni dispersion more active for these reactions?

In the course of finalizing the manuscript we obtained XRD measurements from which the size of Ni crystallites was calculated. Although the crystallite sizes differ significantly in the various catalytic systems, we could not find any meaningful relationships between the dispersion of Ni particles and the activity and stability of our catalytic samples in the studied reactions. For instance, Ni/CaO and Ni/γ-Al2O3 samples exhibit relatively similar yields of carbon products and catalytic activity, while the Ni dispersion for Ni/γ-Al2O3 was twice as high. We cannot exclude the existence of a correlation between the metal particle size and the activity of corresponding catalysts; however, the obtained results did not provide clear evidence for this. The dependence of the catalysts' activity on the reduction degree is more prominent and straightforward. We have added the explanation to the manuscript (page 11).

Moreover, the title of the section 3.1. Activity of catalysts is inadequate because no activity was reported in this section. I suggest to change to 3.1. Catalyst properties.

We are grateful for your remark. We have changed the title of this section to a more appropriate one (page 7).

We trust that our paper is now completely compliant with your recommendations and we are grateful for your valuable suggestions and remarks.

Sincerely,

Artem Kaporov

Author Response

Dear Reviewer,

We are grateful for the supplement to your comment. We have edited your suggested revisions and included them in the manuscript of our paper (page 3).

We trust that our paper is now completely compliant with your recommendations and we are grateful for your valuable suggestions and remarks.

Sincerely,

Artem Kaporov